# Research Progress on the Effects of Selenium on the Growth and Quality of Tea Plants

**DOI:** 10.3390/plants11192491

**Published:** 2022-09-23

**Authors:** Juan Xiang, Shen Rao, Qiangwen Chen, Weiwei Zhang, Shuiyuan Cheng, Xin Cong, Yue Zhang, Xiaoyan Yang, Feng Xu

**Affiliations:** 1College of Horticulture and Gardening, Yangtze University, Jinzhou 434025, China; 2National R&D Center for Se-Rich Agricultural Products Processing, School of Modern Industry for Selenium Science and Engineering, Wuhan Polytechnic University, Wuhan 430023, China; 3Enshi Se-Run Material Engineering Technology Co., Ltd., Enshi 445000, China; 4Henry Fok School of Biology and Agricultural, Shaoguan University, Shaoguan 512005, China

**Keywords:** selenium, *Camellia sinensis* (L.) O. Kuntze, growth, quality

## Abstract

Selenium (Se) is an essential trace element for humans and animals, and it plays an important role in immune regulation and disease prevention. Tea is one of the top three beverages in the world, and it contains active ingredients such as polyphenols, theanine, flavonoids, and volatile substances, which have important health benefits. The tea tree has suitable Se aggregation ability, which can absorb inorganic Se and transform it into safe and effective organic Se through absorption by the human body, thereby improving human immunity and preventing the occurrence of many diseases. Recent studies have proven that 50~100.0 mg/L exogenous Se can promote photosynthesis and absorption of mineral elements in tea trees and increase their biomass. The content of total Se and organic selenides in tea leaves significantly increases and promotes the accumulation of polyphenols, theanine, flavonoids, and volatile secondary metabolites, thereby improving the nutritional quality of tea leaves. This paper summarizes previous research on the effects of exogenous Se treatment on the growth and quality of tea trees to provide a theoretical basis and technical support for the germplasm selection and exploitation of Se-rich tea.

## 1. Introduction

Selenium (Se) is an essential micronutrient for animals to humans, and it is an important component of selenoproteins, glutathione peroxidase (GPx), and thioredoxin reductase (TrxR), which regulate the cellular oxidative stress response through the redox response system. A total of 25 selenoproteins are identified in human cells, which exhibit various biological activities, such as redox signaling, antioxidant defense, and immune response, and they have important functions in antioxidant, immune regulation, and disease prevention. In addition, Se is involved in cell growth, apoptosis and modification of cell signaling systems and transcription factors [1,2]. Se deficiency in the human body can lead to the occurrence of diseases such as Keshan disease, Kaschin-beck disease, and infertility. Given the uneven distribution of selenium resources in the global soil and the Given of selenium-deficient zones, about one billion people are currently facing health risks because of insufficient intake of selenium in their diets, resulting in selenium deficiency. Therefore, scientific and reasonable selenium supplementation through daily diet is essential to maintain human health [3,4]. 

As one of the top three non-alcoholic beverages in the world, tea has received considerable attention from the public for its unique taste and nutritional and health benefits. Studies have shown that long-term tea consumption can reduce serum cholesterol levels and the risk of cardiovascular disease in humans [5]. The tea tree (*Camellia sinensis* (L.) O. Kuntze) has a high-selenium enrichment capacity, converting absorbed inorganic Se into organic Se, only about 8% of Se in tea leaves is inorganic, such as selenite, and the rest of Se is organic such as Se-containing proteins and amino acids. Organic Se is safer and more bioavailable than the inorganic one [6]. The exogenous application of Se can increase the biomass of tea trees while promoting the uptake of mineral elements and inhibiting the uptake of harmful elements. In recent years, numerous studies have shown that exogenous Se fortification can significantly increase the Se content in tea leaves and improve the growth and quality of tea. The review analysis of high-quality articles using the keyword co-occurrence time map [7] showed that the related research mainly focuses on the influence of secondary metabolism in tea on human health, such as increasing antioxidant capacity and reducing the risk of cancer and so on (Appendix A), while the mechanism of exogenous selenium on tea growth and nutritional quality is still insufficient.so there is an urgent need to conduct relevant research to further promote the research on quality selenium tea. In this paper, we review the studies on the effects of exogenous Se on the growth and quality of tea trees and provide theoretical support for the selection and development of Se-rich tea trees.

## 2. Selenium Uptake and Metabolism in Tea Plant

Inorganic Se uptake by plants primarily consists of selenate and selenite, with selenate primarily existing in alkaline-oxidizing environments and selenite primarily existing in acidic and neutral environments. Tea plants can grow and develop in weakly acidic soils, thereby indicating their stronger uptake of selenite [8]. The organic matter in the soil can fix Se, and when the pH of the soil changes from acidic to alkalescence, the elemental Se and selenides in the soil will slowly oxidize to selenate and selenite. Moreover, the divalent Se produced by the decomposition of organically bound Se will also oxidize to selenate and selenite, thereby increasing the effective Se content in the soil [9]. The roots of tea plants are more efficient at selenite uptake than selenate [10]. When selenite is absorbed by the roots of tea plants, it can be accumulated in the roots through phosphate transport proteins and rapidly converted into organoselenium compounds such as selenocysteine (SeCys), selenocystine (SeCys_2_), selenomethionine (SeMet), and methylselenocysteine (MeSeCys) before being transported aboveground (Figure 1). The roots of tea trees take up selenate via sulfate transport proteins and transport it rapidly from the xylem to the aboveground. However, selenate is toxic to plants because it is incompletely transformed in the roots [11]. Previous studies have hypothesized that higher selenate concentrations in plant cells can reduce plant water potential, resulting in the rapid translocation of selenate to the aboveground. Selenate is converted into organoselenides, such as SeCys, SeMet, and MeSeCys, in the leaves through the sulfate metabolism pathway. These organoselenides have oxidative, anti-inflammatory, and other biological activities, and they are important components of the nutritional quality of tea leaves [12].

Tea plant leaves have not evolved a specific structure to absorb Se. Studies have shown that under the conditions of open leaf stomata, the absorption rate of external solutes by plant leaves can be significantly increased [13]. However, plants have evolved protective structures such as waxy layers and cuticular layers to protect the fragile leaf tissue; for example, the upper and lower epidermal layers of tea tree leaves are covered with a cuticular layer composed of keratin and wax, which hinders the absorption of Se in the leaves [14]. Research shows that plant leaves absorb Se as selenate or selenite. They can also absorb organic Se such as SeCys and SeMet but not insoluble elemental forms of Se (Se0) or Se metal compounds [15]. Consequently, most of the inorganic Se absorbed by the leaves is converted to organic Se, primarily in the form of soluble proteins [16].

The uptake and metabolism of selenite and selenate in rice (*Oryza sativa*) and tobacco (*Nicotiana tabacum*) is similar to that of tea trees. The selenite is absorbed and metabolized in the underground part of the plant and then transferred to the aboveground part. However, selenate is absorbed by roots and then transported to the aboveground part, which is metabolized and converted into organic selenium in leaves. Studies on wheat also showed similar results [17,18]. At present, the research on the absorption, metabolism, and regulation mechanism of selenium in tea leaves is still insufficient, which needs to be further explored in future work. However, studies on the uptake and metabolism of Se by tea leaves remain insufficient and need to be further explored at a later stage.

## 3. Effect of Selenium on Tea Plant Growth

### 3.1. Effect of Selenium on Tea Plant Biomass

The health functions and pharmacological mechanisms of tea are constantly being explored and utilized by modern science. Tea has become the world’s largest cultivated and most widely consumed beverage crop to improve tea production and promote the early sprouting and expansion of tea buds; thus, the spring tea early market can effectively improve the economic benefits of the market. Investigation shows that exogenous application of Se can significantly increase the yield of tea leaves and promote early germination of tea plants [19]. Huang et al. showed that foliar application of Se could effectively increase the 100-bud weight (weight of one 100 growing buds) of tea trees, and the 100-bud weight gradually increased with the spraying dosage when the concentration of exogenous Se was controlled at 30 and 50 mg/kg, and gradually decreased with the spraying dosage at 100 mg/kg. This result indicates that the tea yield can be significantly increased under the appropriate concentration of Se treatment, but the concentration of Se is excessively high to reduce the 100-bud weight of tea plants [20]. Wu et al. used the tea variety “Baiye No. 1” as the test material for foliar spraying of nano-Se and found that the yield of fresh tea leaves was significantly improved (Table 1). In particular, when the foliar spraying of nano-Se was applied at a concentration of 13.5 g/hm^2^, the photosynthetic performance and yield of tea trees could be significantly improved. This result could be attributed to the fact that the net photosynthetic rate, transpiration rate, and stomatal conductance of tea trees were enhanced, and the intercellular CO_2_ con-centration was reduced at this Se concentration, resulting in the greatest increase in photosynthetic performance [21]. Xu et al. also found that Se-containing biologics at a concentration of 100 mg/L and at a rate of 50 g/ha significantly promoted earlier germination of tea plants in early spring and increased the yield of high-grade (Table 1), high-quality tea leaves by a factor of two in early spring [22]. Se treatment not only increases the net photosynthetic rate of tea leaves under low-temperature stress to stabilize plant photosynthesis and membrane systems but also improves the cold tolerance of tea plants. Compared to the CK treatment, the Fv/Fm value of tea leaves increased by 10.63%, and the photochemical quenching value increased by 39.45% under 2 mg/mL exogenous selenium treatment [23]. Therefore, the appropriate Se concentration not only enables tea trees to germinate earlier but also significantly increases the biomass of tea trees and improves tea production.

### 3.2. Effect of Selenium on the Uptake of Mineral Elements by Tea Plant

Tea leaves are rich in a variety of mineral elements, and they have not only an important role in the growth and development of tea trees but are also an important expression of the nutritional value of tea. Based on relevant studies, the exogenous application of Se can have different effects on the uptake of mineral elements in tea plants. Foliar spraying of low concentrations (5.0~50.0 mg/L) of exogenous Se (sodium selenite and sodium selenate) can increase the Zn, K, Ca, and Mg content of tea leaves to some extent, and 50.0 mg/L Se treatment has a significant effect on the Fe content of leaves (Table 1) [28]. In the case of Se deficiency, the fluorine (F) content in tea leaves and roots increased significantly with the increasing exogenous Se concentration. In the case of Se sufficiency, the Fe, Ca, and Al content in tea leaves increased, the Se and Mg content in leaves and roots increased significantly, and the total F, water-soluble F, and malondialdehyde (MDA) content decreased significantly [24,29,30]. There are few reports on the effect of selenium on the absorption of mineral elements in tea plants. The above studies showed that exogenous application of selenium could increase the accumulation of mineral elements such as Zn, Mg, and Fe but inhibit the absorption of F by roots. However, the absorption of other mineral elements has not been reported, so it needs to be verified by subsequent experiments.

## 4. Effects of Selenium on Tea Plant Quality

### 4.1. Effect of Exogenous Selenium on the Selenium Content of Tea Plants

The exogenous application of Se is an effective way to increase the Se content of tea and produce Se-rich tea. Related studies have found little variation in the year-to-year background values of Se content in different parts of tea trees without Se application [31].

The total selenium content of tea trees at suitable selenium concentrations (5.0~100.0 mg/L) showed an increasing trend and showed significant differences at 25 mg/L, but high concentrations of selenium (150 mg/L) would have toxic effects on the growth of tea trees (Table 1); in the third year after soil application of sodium selenite, the selenium content of all parts of tea trees was significantly increased, especially the root selenium content, and the roots of tea trees would absorb selenium, and a large amount of organic selenium was formed rapidly in its roots, which indicated that root application of selenium could effectively increase the selenium content of tea trees and showed an increasing trend with the increase in selenium application concentration [28,32]. This phenomenon may be due to the rapid formation of large amounts of organic Se, including SeMet and MeSeCys, in the roots of tea trees when selenite is absorbed by the roots as compared with sodium selenate [33,34]. 

Wang et al. used different concentrations (0, 0.15, 0.3, 1.5, 3, 5, 8 mg/L) of sodium selenate in hydroponic trials on annual tea seedlings and found that the total Se content of both the roots and new parts of the tea seedlings increased with increasing Se concentration, but the selenium concentration above 3 mg/L had an inhibitory effect on the growth of tea seedlings [35]. In addition, the aboveground content was significantly higher than the belowground content, indicating that tea tree roots can absorb sodium selenate and transfer it to stems and leaves, thereby increasing the aboveground Se content [25]. The total Se content of tea trees is also related to the harvesting time. Hu et al. showed that it is 60 mg of Se/L and a rate of 75 g/ha used fertilizer of selenate and fertilizer of selenite at a concentration of was sprayed onto old leaves of the tea trees. It is appropriate to harvest tea trees within 10–20 days after Se application is appropriate, showing that the Se content of tea leaves may decrease as the growth time increases [19]. Based on these studies, the production of Se-rich tea should consider tea tree varieties, Se application methods, and the choice of exogenous Se. Suitable exogenous Se application can promote tea tree growth and significantly increase the total Se content of tea trees. 

### 4.2. Effect of Exogenous Selenium on Selenium Forms in Tea Plants

Se exists as organic and inorganic forms in tea leaves, but some low-molecular-weight organic Se and inorganic Se can be extracted into tea infusion. Hence, knowledge of Se speciation, particularly its inorganic form with high toxicity and its distribution in Se-rich tea leaves and tea infusion, are important to food production, nutrition, and safety [36]. Tea plants have a strong Se accumulation capacity, and they can convert inorganic Se (SeO_3_^2−^ or SeO_4_^2−^) into safe organic Se, such as SeCys_2_ and SeMet [34]. Qin et al. found that 6 to 36 days after foliar spraying, the Se in the leaves was primarily in the organic form, with the organic Se content accounting for 93.3% to 96.6% of the total Se content, which shows a small variation, indicating that 6 days after foliar spraying, most of the inorganic Se had been converted into organic Se [28]. Most of the Se-rich tea leaves are organic Se, and only about 8% of the Se is in the inorganic form. Moreover, the organic form of Se is about 30–35% in tea protein, and the rest exists in a free state in the solution [6].

After Se is absorbed by the tea plants, it is transferred to the ground and synthesized into other Se-containing substances. The organic Se in tea is primarily composed of macromolecular Se and seleno-substituted amino acids and derivatives of small-molecule selenides. Small-molecule selenides include SeMet, SeCys_2_, SeCys, and MeSeCys (Figure 2), and seleno-substituted amino acids are the main source of Se in the human daily diet [37,38]. Tea trees have 92% of Se as organic Se, such as Se-containing proteins and amino acids, and organic Se is safer and more bioavailable than inorganic Se. Generally, the content of organic Se in tea increases with the increase in total Se content in tea; therefore, combined with previous studies, developing an optimal plan including Se source, Se application concentration, Se application season, and harvesting time is an important way to increase the total and organic Se content in tea.

### 4.3. Effect of Exogenous Selenium on Tea Polyphenols in Tea Leaves

Tea polyphenols is the general name of polyphenols in tea, including flavanols, flavonoids, anthocyanins, leucoanthocyanidins, phenolic acid, and depside. Polyphenols cause the astringent and bitter taste of tea soup, and they are important nutritional elements in tea [39,40]. Tea polyphenols have a variety of biological effects. They are considered to be immune-enhancing nutrients that can effectively regulate the body’s innate immune response, even against the novel coronavirus disease COVID-19 [41]. Li et al. showed that 10 mg/L Se nanoparticles significantly increased the polyphenol content of green tea, but 20 mg/L Se nanoparticles inhibited the accumulation of tea polyphenols [42]. In addition, Se treatment reduced the tea polyphenol content in early summer tea leaves compared with tea leaves without Se treatment, and the content did not change significantly in autumn tea leaves. Moreover, common green tea harvested in early spring had the highest tea polyphenol content among the three seasons at 240.92 g/kg (Table 1), which causes the high astringency of early spring green tea broth [19,26]. Wang et al. reported that the quality of finished tea after organic Se spraying treatment found that the longer the Se treatment on the leaf surface of tea trees, the less the tea polyphenol content, and the change was negatively correlated, indicating that Se application to tea trees can weaken the bitterness and astringency in the tea broth because of the reduced ratio of tea polyphenols to amino acids in tea leaves [43]. The abovementioned study shows that the effect of exogenous Se on the polyphenol content of tea leaves is related to the season of Se spraying and the time of harvesting.

During the storage of the finished tea, the polyphenol content in Se-sweetened green tea is higher than that of regular green tea [27]. Thus, the organic Se in tea leaves and the polyphenols in tea leaves may have synergistic effects; that is, high-Se tea has higher total phenolic content and antioxidant activity than green tea containing normal amounts of Se, and the polyphenolic compounds in green tea extracts are effective free radical scavengers [44,45].

### 4.4. Effect of Exogenous Selenium on Flavonoids in Tea Leaves

The main polyphenols in tea include flavonoids. Diet has six major classes of flavonoids, including flavonols, flavones, flavanols, flavanones, anthocyanins, and isoflavones. The most common subclasses of flavonoids in tea are flavanols (primarily catechins and quercetin) [46]. Flavonoid compounds have a preventive and therapeutic effect on cancer and cardiovascular diseases [47]. Barreca et al. isolated a Se-containing flavonoid from Se-rich green tea by high-speed counter-current chromatography (HSCCC) and characterized it by UHPLC-Q-Orbitrap, FT-IR, and NMR, showing the suitable solubility of the Se-containing flavonoid. It also has a superior antioxidant activity to flavonoids, effectively inhibiting the production of nitric oxide (NO) and playing an important role in the process of oxidative stress and severe inflammatory damage in the body. In general, flavonoids are planar molecules with low solubility caused by their tight intermolecular packing, which hinders their entry into cells, and they are also difficult to utilize. Studies have confirmed that the introduction of selenite ions breaks the planar structure of flavonoids and reduces the intermolecular attraction, facilitating the entry of water molecules and cellular uptake, which results in stronger pharmacological effects of Se-containing flavonoids compared with common flavonoids [48,49]. Hu et al. found that tea tree flavonoid biosynthesis genes were significantly enriched in the sodium selenate-treated group, indicating that Se application could promote the synthesis of flavonoids [50]. In addition, bHLH-like transcription factors can interact with MYB transcription factors to participate in the accumulation of secondary metabolites such as flavonoids and anthocyanins under Se treatment [51]. Se supplementation could promote secondary metabolism in tea, thereby increasing the accumulation of flavonoids (apigenin, kaempferol, quercetin, myricetin, and rutin) [42]. At present, most studies focus on the effect of exogenous selenium on the content of secondary metabolites in tea, but the molecular mechanism is not clear. At present, only a few pieces of literature have reported that the intermediate products of the selenium metabolic pathway in plants can provide precursors for the synthesis of important secondary metabolites such as flavonoids and phenolic acids [52]. The mechanism of selenium affecting the accumulation of secondary metabolites is also the focus of future research.

### 4.5. Effect of Exogenous Selenium on Amino Acids in Tea Leaves

The content and composition of free amino acids have a significant effect on the taste, color, aroma, and freshness of the tea broth. Theanine, a special chemical component of tea, accounts for 50% of the free amino acids, and it has high sweetness and freshness, which not only constitutes the freshness of the tea broth but also reduces the bitterness of the tea [53]. During foliar applications in autumn, selenite and selenate increased the Se, total amino acid, and vitamin content of tea [26]. Moreover, compared with ordinary green tea, the content of thiamin pyrophosphate in Se-rich green tea decreases as the particle size of green tea decreases, whereas the content of amino acids remains stable [54]. Liu et al. found that N assimilation could be regulated through the glutamine synthetase-glutamate synthase (GS-GOGAT) pathway, and its products could serve as signaling molecules or precursors that could further regulate primary and secondary metabolism in plants [55]. The nano-Se treatment enhanced the activity of GOGAT and GS, and arginine, glutamic acid, proline, and theanine were synthesized through the GS-GOGAT cycle. In addition, aspartic acid and serine were higher in treated samples than in the untreated group, in which foliar Se nano-spray at a concentration of 10 mg/L significantly increased glutamic acid (55.4%), aspartic acid (45.5%), and serine (37.9%) levels [42]. The abovementioned studies suggest that suitable Se treatment facilitates the accumulation of theanine in tea leaves, which may be closely related to amino acid, carbohydrate, and secondary metabolic synthesis [56].

### 4.6. Effect of Exogenous Selenium on Other Nutrients in Tea Leaves

Tea also contains the alkaloids methylxanthine, that is, caffeine, and its intermediates theophylline and theobromine. They have the structure of purine nucleosides, which enables them to serve as competitive inhibitors of adenosine receptors. This G protein-coupled receptor is widely present in immune cells in different isoforms, and it plays a key role in the regulation of the innate immune response [57]. Caffeine and tea polysaccharides (TPS) are important bioactive compounds in tea, and they play an important role in regulating metabolic syndrome. Caffeine serves as an antagonist of adenosine receptors, blocking A1 adenosine receptors in adipocytes, promoting lipolysis in adipocytes, and reducing systemic fat to regulate metabolic syndrome in individuals with diet-induced obesity. TPS serves as a carbon source for the intestinal microbiota to regulate metabolic syndrome by controlling the structure and fermentation of the intestinal microbiota [6]. TPS is primarily extracted from tea leaves, and they have a variety of medicinal properties that are beneficial to human health, such as anti-oxidants, anti-diabetic, and anti-inflammatory. To date, less attention has been paid to artificial Se TPS, and almost no studies on the structural properties and activity of artificial Se TPS have been reported in the past 20 years, which has greatly hindered the progress of research on artificial Se TPS. The application of artificial Se significantly increased the total Se content in TPS, and 2.45% glucosamine was found in Se-enriched TPS, indicating that new monosaccharide components may be synthesized by artificial selenization TPS [21]. In addition, the exogenous application of Se significantly increased the vitamin C content in tea leaves. In another previous study, the Se-containing polysaccharide Se-ZGTP-I extracted from Ziyang green tea significantly inhibited keloid fibroblast development in vitro by inducing apoptosis and inhibiting type I collagen synthesis [58].

### 4.7. Effect of Exogenous Selenium on Aroma Composition in Tea Leaves

Aromatic substances in tea are also known as “volatile aromatic components,” which is a general term for volatile substances in tea leaves. They contain a mixture of many volatile substances with different properties, minimal content, and significant differences, which are directly related to the sensory quality and physiological health functions of tea leaves [59]. Li characterized and identified volatile flavor substances in green tea treated with Se nanoparticles and found a variety of aroma components, including alcohols, terpenes, aldehydes, esters, ketones, acids, aromatic hydrocarbon furans, thiols, pyrazines, and sulfides. Foliar spraying of Se nanoparticles increased the content of aroma components in tea [42]. The determination of volatile aroma components of green tea in different treatment groups using GC-IMS revealed an increase in methyl salicylate, acetone, ethyl acetate, toluene, dimethyl disulfide, 3-methylbutanoic acid, ethanol, propionic acid, 2-heptanone, lauryl, benzaldehyde-M, and p-xylene levels in Se-treated samples, whereas the levels of butyraldehyde, isobutyl acetate, 3-methylbutyraldehyde, 3-methylbutanol, linalool, and 2-ethylfuran decreased. In general, laurylenes have a floral and sweet aroma; ethyl acetate and 2-heptanone have a strong fruity aroma that creates the distinctive flavor of green tea; acetone at low concentrations gives off a light aroma and promotes the harmonization of ester aroma; methyl salicylate and dimethyl disulfide contribute to the chestnut-like aroma of green tea [60]. Benzaldehyde is a volatile substance derived from phenylalanine in tea, and it has an odor characteristic of bitter almonds [61]. In addition, the foliar spraying of Se nanoparticles increased the content of linalool and 3-methylbutyraldehyde, which also create the chestnut aroma in green tea. Consequently, the aroma of tea treated with Se nanoparticles was more intense, and the key odor substances contributed to the appearance of chestnut aroma in the finished tea through synergistic and indirect effects [62].

## 5. Conclusions and Perspectives

Se is an essential trace element in the human body, and drinking Se-rich tea is a convenient and effective way to supplement Se, which is important for maintaining human health. At present, research on Se-rich tea focuses on two aspects of Se-rich cultivation and quality research, which has made certain research results. Recent studies have proven that an appropriate amount of exogenous Se can promote photosynthesis and mineral element absorption in tea trees, inhibit the absorption of harmful elements, and improve biomass; meanwhile, it significantly increases the organic Se content of tea leaves and promotes the accumulation of amino acids, tea polyphenols, flavonoids, and volatile secondary metabolites, thereby improving the quality of tea leaves (Figure 3). However, many shortcomings are still found in related research.

Although low-Se tea plantations can improve the yield of Se-rich tea through exogenous Se supplementation, the differences in tea tree species resources and growth environment will lead to instability in the yield and nutritional quality of Se-rich tea. The transport and metabolic molecular mechanism of different Se sources in tea trees are not comprehensively studied, and the understanding of the mechanism of Se concentration in tea trees remains insufficient, which hinders the selection and breeding of Se-rich tea tree germplasm resources and the development of large-scale cultivation. Therefore, future research must address the abovementioned issues in a more systematic and comprehensive manner, particularly germplasm resource mining and cultivation control measures, to simultaneously improve the quality and organic Se content of tea leaves.

## Figures and Tables

**Figure 1 plants-11-02491-f001:**
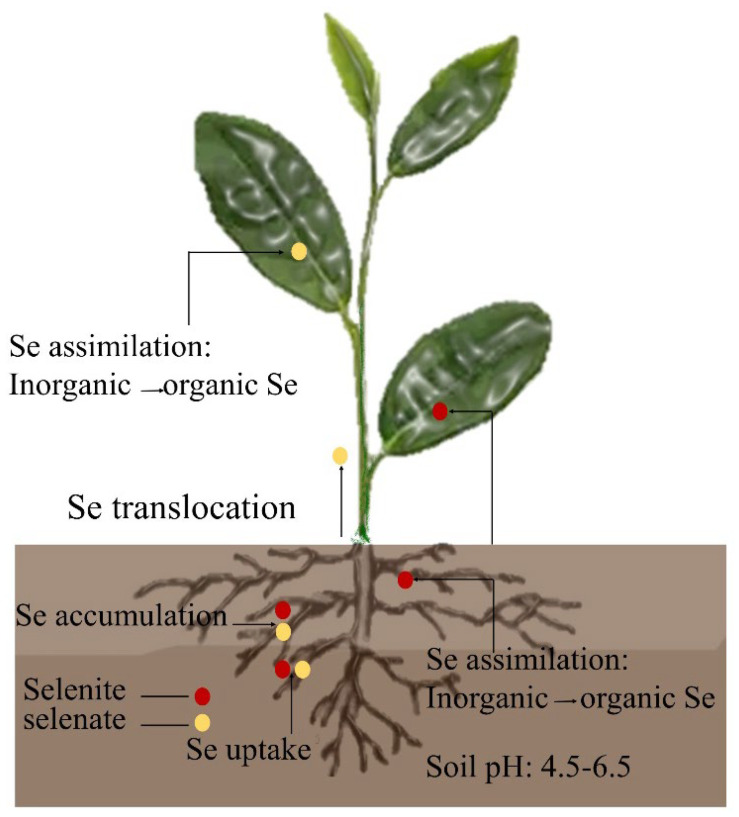
Uptake and transport of selenate and selenite by tea plants.

**Figure 2 plants-11-02491-f002:**
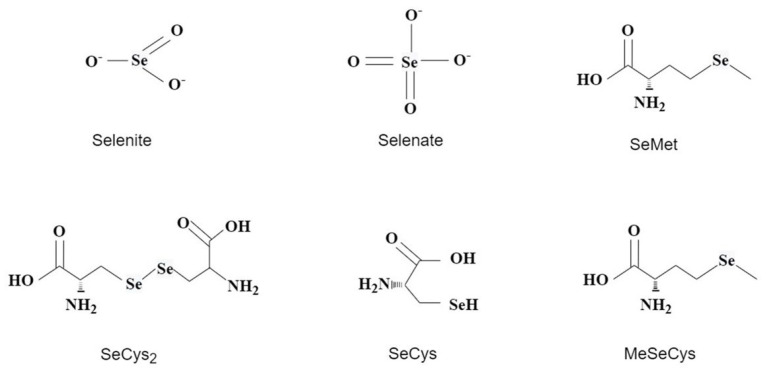
Main inorganic selenium and organic selenium species in tea plants.

**Figure 3 plants-11-02491-f003:**
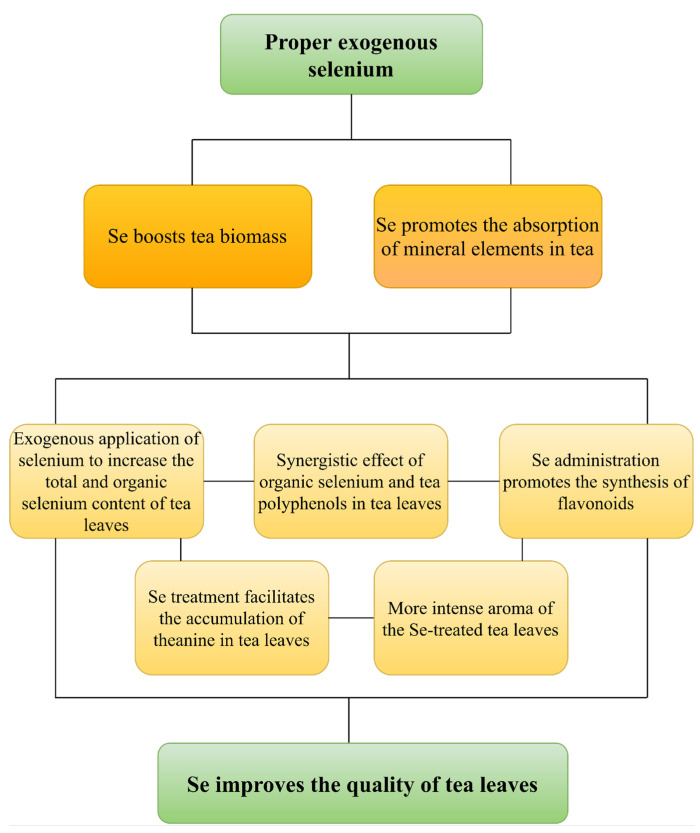
Proper exogenous selenium improves the biomass and nutritional quality of tea plants.

**Table 1 plants-11-02491-t001:** Tea plant: different species, selenium treatment (Se source, dose, type of treatment), and total Se content and nutritional substance.

Tea Species	Se Source	Dose	Type of Treatment	Se Content (DW)	Increased Nutrient Content	References
Early Spring Green Tea	selenite and selenate fertilizer	60 mg /L	Field foliar spraying	7.5–10.6 mg/kg	Amino acid; vitamin C FW	[20]
Wu Niuzao	organic Se	100 mg/kg	Field foliar spraying	4.72 mg/kg	Organic selenium; polyphenol; caffeine DW	[21]
Baiye No.1	Nano-Se	13.5 g/hm^2^	Field foliar spraying	NA	Significant increase in chlorophyll content FW	[22]
Early Summer Green Tea	Se-enriched fertilizer	100 mg/kg	Field foliar spraying	5.895 mg/kg	Vitamin C FW; tea polyphenol DW	[23]
Guilv No.1	sodium selenite	100 mg/L	Field foliar spraying	15.88 mg/kg	Organic selenium; Zn, K, Fe, Ca, and Mg DW	[24]
Qiancha 601	sodium selenate	0.3 mg/L	hydroponics	≥0.25 mg/kg	Chlorophyll FW; tea polyphenol DW	[25]
Zhongcha 108	Nano-Se	10 mg/L	Field foliar spraying	1–1.5 mg/kg	Tea polyphenol; flavonoids; caffeine DW; amino acid chlorophyll FW	[26]
Tea No. 12	organic Se	750–2100 g/hm^2^	Field foliar spraying	0.344–1.111 mg/kg	Tea polyphenol; caffeine DW	[27]

Note: NA, not analyzed; DW, dry matter; FW, fresh weight.

## Data Availability

Not applicable.

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
