# Peer review of "Research Progress on the Effects of Selenium on the Growth and Quality of Tea Plants"

_plants, 2022, doi:10.3390/plants11192491_

Round 1
Reviewer 1 Report
1- Abstract needs some statistics and/or quantifying results
2- Introduction section is too weak and must be improved by highlighting the hypothesis, and the novelty of the study.
3- The recommended doses of Se should be explained under different environments with related references.
4- This is a review paper, but I don't see how authors prepared that!! The best literature review should reveal, assess, and structure the relevant literature on the intended topic, as well as combine it with a critical analysis of various arguments in the literature. Authors must add an additional section called "Survey Methods" and add a flowchart for methods used in the Review. For help, you can use this paper (http://doi.org/10.7717/peerj.13674)
5- Findings for the review look weak, you must add a Table or Figure for major findings from the literature, and what are the limitations and future directions.
Reviewer 2 Report
This manuscript entitled "Research progress on the effects of selenium on the growth and 2 quality of tea plants" was reviewed carefully. I think it is an interesting review, and in a good written. I suggested it could be acceptable after some minor revision.
Lines 116, 131, 150, 158, 177, 208, 236, please supplement the necessary feference.
1. it was an interesting manuscript.
2. it gave a clear description for the effects of seleniunm on tea plants.
3. the paper was well written, and easy to read.
Reviewer 3 Report
This manuscript tries to summarize the recent studies of Se in tea plants. However, the writing is very confusing and needs to reorganize. The figures in the manuscript are very poor and lack key information, which needs to be improved.
1. After selenite is converted into organoselenium in tea roots, whether the organoselenium will be transported aboveground or not? And according to the note below figure 1, it showed the organoselenium would be transported aboveground, please also show it in Figure 1.
2. Please add some discussion on the studies of selenium uptake mechanism in other plants at the end of part 2. Selenium uptake and metabolism in tea plant.
3. Lines 116, 131,150, 158, 177, and 208 there is an error in the reference. Please check it.
4. Line 108 cites the reference as the full name Wu Huanhuan while line 116 is the family name, please check it in the whole manuscript carefully.
5. The content of Lines 141-147 is similar to the content of lines 102-108. Please reorganized these two parts.
6. What’s the detailed relationship between flavonoids, and flavonols with Se in figure 2B? Please show it more clearly. In the manuscript, Se also showed the effect on caffeine, theanine, aroma, etc. So please also add this information in figure 2.
7. When the exogenous application of Se in large amounts it did not improve the quality of tea leaves, the conclusion of Figure 3 should be modified.
Round 2
Reviewer 1 Report
Could be accepted now.
Reviewer 3 Report
It could be accepted.